# Genetic Characterization of Endometriosis Patients: Review of the Literature and a Prospective Cohort Study on a Mediterranean Population

**DOI:** 10.3390/ijms21051765

**Published:** 2020-03-04

**Authors:** Stefano Angioni, Maurizio Nicola D’Alterio, Alessandra Coiana, Franco Anni, Stefano Gessa, Danilo Deiana

**Affiliations:** 1Department of Surgical Science, University of Cagliari, Cittadella Universitaria Blocco I, Asse Didattico Medicna P2, Monserrato, 09042 Cagliari, Italy; danideiana82@gmail.com; 2Department of Medical Science and Public Health, University of Cagliari, Laboratory of Genetics and Genomics, Pediatric Hospital Microcitemico “A. Cao”, Via Edward Jenner, 09121 Cagliari, Italy; acoiana@unica.it; 3Department of Medical Science and Public Health, University of Cagliari, Cittadella Universitaria di Monserrato, Asse Didattico E, Monserrato, 09042 Cagliari, Italy; francoanni@gmail.com; 4Laboratory of Molecular Genetics, Service of Forensic Medicine, AOU Cagliari, Via Ospedale 54, 09124 Cagliari, Italy; stefagessa@libero.it

**Keywords:** endometriosis, genetic polymorphisms, SNPs, pathogenesis, aetiology, Mediterranean population, Sardinian population

## Abstract

The pathogenesis of endometriosis is unknown, but some evidence supports a genetic predisposition. The purpose of this study was to evaluate the recent literature on the genetic characterization of women affected by endometriosis and to evaluate the influence of polymorphisms of the wingless-type mammalian mouse tumour virus integration site family member 4 (WNT4), vezatin (VEZT), and follicle stimulating hormone beta polypeptide (FSHB) genes, already known to be involved in molecular mechanisms associated with the proliferation and development of endometriotic lesions in the Sardinian population. **Materials and Methods:** In order to provide a comprehensive and systematic tool for those approaching the genetics of endometriosis, the most cited review, observational, cohort and case-control studies that have evaluated the genetics of endometriosis in the last 20 years were collected. Moreover, 72 women were recruited for a molecular biology analysis of whole-blood samples—41 patients affected by symptomatic endometriosis and 31 controls. The molecular typing of three single nucleotide polymorphisms (SNPs) was evaluated in patients and controls: rs7521902, rs10859871 and rs11031006, mapped respectively in the WNT4, VEZT and FSHB genes. In this work, the frequency of alleles, genotypes and haplotypes of these SNPs in Sardinian women is described. **Results:** From the initial search, a total of 73 articles were chosen. An analysis of the literature showed that in endometriosis pathogenesis, the contribution of genetics has been well supported by many studies. The frequency of genotypes observed in the groups of the study population of 72 women was globally coherent with the law of the Hardy–Weinberg equilibrium. For the SNP rs11031006 (FSHB), the endometriosis group did not show an increase in genotypic or allelic frequency due to this polymorphism compared to the control group (*p* = 0.9999, odds ratio (OR) = 0.000, 95% confidence interval (CI), 0.000–15.000 and *p* = 0.731, OR = 1639, 95% CI, 0.39–683, respectively, for the heterozygous genotype and the polymorphic minor allele). For the SNP rs10859871 (VEZT), we found a significant difference in the frequency of the homozygous genotype in the control group compared to the affected women (*p* = 0.0111, OR = 0.0602, 95% CI, 0.005–0.501). For the SNP rs7521902 (WNT4), no increase in genotypic or allelic frequency between the two groups was shown (*p* = 0.3088, OR = 0.4133, 95% CI, 0.10–1.8 and *p* = 0.3297, OR = 2257, 95% CI, 0.55–914, respectively, for the heterozygous genotype and the polymorphic minor allele). **Conclusion:** An analysis of recent publications on the genetics of endometriosis showed a discrepancy in the results obtained in different populations. In the Sardinian population, the results obtained do not show a significant association between the investigated variants of the genes and a greater risk of developing endometriosis, although several other studies in the literature have shown the opposite. Anyway, the data underline the importance of evaluating genetic variants in different populations. In fact, in different ethnic groups, it is possible that specific risk alleles could act differently in the pathogenesis of the disease.

## 1. Introduction

Endometriosis is a chronic disease of the reproductive age with a prevalence of 5%, reaching its peak between 25 and 35 years of age [1]. The scientific community agrees in recognizing a multifactorial aetiology of endometriosis, with possible genetic, hormonal, immunological and environmental factors as causes. Studies on family aggregation and twins have emphasized the genetic component, demonstrating how predisposition generated by certain susceptibility genes plays an important role in the development, maintenance and recurrence of the disease [2,3,4,5]. Nonetheless, studies on the genetics of endometriosis are complicated by various factors: the phenotypic heterogeneity of the disease; the still unknown prevalence in the population, burdened by the absence of registries and diagnostic underestimation; the invasiveness of diagnostic methods; and various co-morbidities that can generate bias [3]. In the field of research on the genetic basis of endometriosis, Simpson is considered a pioneer. In 1980, he verified in a sample of 123 women with histological diagnosis of endometriosis that 6.9% of first-degree relatives (mother and sisters) were affected, while the disease prevalence in controls (first-degree female relatives of the corresponding “husbands”) was less than 1% [6]. In 1999, Treloar’s work on an Australian population demonstrated a concordance ratio of 2:1 between monozygotic and dizygotic twins and a correspondent genetic risk of 2.34 to affect a sister. The study results show that 51% of genetic influence is responsible for developing endometriosis [7]. There is therefore enough evidence on how endometriosis is clearly heritable, although in what manner is not yet clear. The increased genetic risk in first-degree relatives (5–8%) suggests polygenic and multifactorial inheritance rather than monogenic. However, this recurrence risk is higher than the expected risk for a polygenic pathology (2–5%). The other, more likely, hypothesis is that phenotypic heterogeneity reflects genetic heterogeneity and that therefore not all forms of endometriosis are the same disease; in fact, some forms of endometriosis, due to their characteristics, behave almost like Mendelian pathologies [2]. The aim of this study was to evaluate the recent literature on the genetic characterization of women affected by endometriosis and to evaluate the influence of polymorphisms of the WNT4, VEZT and FSHB genes, known to be involved in molecular mechanisms associated with proliferation and development of endometriotic lesions in a particular Mediterranean population, the Sardinian population. The people of the Mediterranean island of Sardinia are particularly well suited for genetic studies, as is evident from a number of successes in complex trait and disease mapping [8].

The wingless-type mammalian mouse tumour virus integration site family member 4 (WNT4) gene is positioned on chromosome 1p36.23-p35 and codifies a protein which is essential in developing the female reproductive system [9,10,11,12,13,14]. It critically regulates the appropriate postnatal uterine maturation, as well as ovarian antral follicle growth [10]. The WNT class is an extensive group of secreted glycoproteins, codified through 19 different genes implicated in the WNT signalling pathway [9]. WNT-mediated signal transduction pathways address the specific mobilization of groups of genes which are responsible for managing several cellular responses, including cell growth, differentiation, movement, migration, polarity, cell survival and immune response [9]. A study published by Jordan et al. [15] demonstrated that WNT4 is the first signalling molecule causing the chain of events which ends with sex determination, through local secernment of growth factors. Imperfections in WNT4 activity play a role in the development of three important organs deriving from the primordial urogenital ridge—the kidneys, adrenal glands and gonads [9]. This may demonstrate the significant position of WNT4 at an early embryological stage of development. The loss of WNT4 in knockout mice determines the total absence of the Mullerian duct and its derivatives [11]. Apart from being crucial for epithelial–stromal cell communication in the endometrium, WNT signalling is likely important for endometrial maturation and differentiation and embryonic implantation [16]. An association between endometriosis and markers located in or near WNT4 has been highlighted in a number of extensive studies on gene mapping [17,18]. The expression of WNT4 has also been detected at the level of the peritoneum, leading to the consideration of a possible metaplastic hypothesis in promoting the transformation of peritoneal cells into endometriosic cells, through pathways with a role in the development of the female genital tract [12]. Pagliardini et al. demonstrated that a single nucleotide polymorphism (SNP), rs7521902, located 21 kb up/downstream of the WNT4 region, has a susceptibility locus for endometriosis. The functional significance of this SNP in endometriosis remains to be explained [10]. While the SNP rs7521902 was connected to endometriosis susceptibility in British, Australian, Italian and Japanese women [10,18,19], in Belgian [13] and Brazilian women this association was not found [9]. The different genetic backgrounds of the cohorts may be identified as the reason for the lack of association between this polymorphism and endometriosis.

The vezatin (VEZT) gene is located on chromosome 12 locus 12q22; it codifies vezatin, a significant element of the cadherin–catenin complex, which plays a crucial role in the formation and sustenance of adherent joints [20,21,22,23,24]. According to Kussel-Andermann et al., vezatin was found to be a plasma membrane component with a short extracellular domain, a transmembrane domain and an extended intracellular domain. Its intracellular domain connects to myosin VIIA as part of the adherens junctional complex in epithelial cells [21]. Furthermore, several studies on co-immunoprecipitation showed that the system between vezatin and myosin VIIA is able to interact with the system between E-cadherin and catenin, although the specificity of this interaction remains to be determined [21,22]. Also, VEZT is fundamental for implantation; embryos from mice with silenced VEZT cannot develop after the blastocyst stage, because of a loss of adhesion between cells [25]. It has been highlighted that VEZT protein is extensively expressed in human endometrium and myometrium. During the secretory phase of the menstrual cycle, VEZT expression increases in the glandular epithelium in a significant way. The mRNA expression of adherens junction members (E-cadherin and A- and B-catenin) is also enhanced in the secretory phase with respect to the proliferative phase. This indicates that progesterone could be responsible for activating cell-to-cell adhesion [20]. The VEZT promoter does not contain a reaction point for the progesterone receptor (PR), but it contains a nuclear factor kappa B (NF-kB) binding site. As a pro-inflammatory transcription factor, NF-kB is involved in the pathogenesis of endometriosis, showing cycle control in the endometrium and reciprocal management with PR [26]. Therefore, variations in VEZT levels in endometrial glandular cells are likely to occur in response to the dynamic oscillation in progesterone and associated NF-kB changes [20]. Considering the studied physiological roles of VEZT, its potential for a functional role in endometriosis is a likely option, since VEZT has been demonstrated to be upregulated in ectopic endometrium with respect to eutopic endometrium in patients suffering from endometriosis [20,23,24].

The follicle-stimulating hormone beta polypeptide (FSHB) gene, positioned on chromosome 11 locus 11p14.1, codifies subunit b of the hormone-specific follicle-stimulating hormone (FSH), with a crucial role in the growth of ovarian follicles and production of oestrogens [27,28,29]. Recently, some evidence for an association between endometriosis and SNPs of FSHB was reported in independent targets from the UK Biobank, firmly supporting this result [28]. FSH and luteinizing hormone (LH) are related gonadotropin hormones sharing the same alpha subunit. A connection between these SNPs on chromosome 11 with concentrations of both hormones indicates a common mechanism of regulation, with both being key elements in managing follicle development in the ovary, influencing oestradiol release during the proliferative phase of the cycle and contributing to a role for oestradiol in endometriosis risk [27]. Data from the ENCODE project [30] show that the SNP rs11031006 modifies the sequence of 11 protein-binding motifs, including that of oestrogen receptor α, with a possible effect on hormonal feedback inhibition. Recently, allele G of this SNP has been proven to be significantly associated with higher levels of serum FSH [29].

### 1.1. Review of the Literature

This review aims to provide a comprehensive and systematic tool for those approaching the genetics of endometriosis. Computerized literature searches were conducted using the Medline/PubMed database and a manual search of relevant and frequently cited publications in the English language from 1999 to 2019. Additional articles were identified by manually searching references from the retrieved eligible articles. Keywords included: endometriosis combined with genes, genetics, SNPs, genome-wide association study (GWAS), WNT4, VEZT, FSHB, next-generation sequencing (NGS) and epigenetics. Review, observational, cohort and case-control studies that evaluated the genetics of endometriosis are herein described.

### 1.2. Cohort Study

The study was carried out through molecular typing of the following single nucleotide polymorphisms (SNPs): rs7521902, rs10859871 and rs11031006, mapped, respectively, in the WNT4, VEZT and FSHB genes. In this work, we set out to describe the frequency of alleles and genotypes of these SNPs among Sardinian women, and to evaluate their impact on the susceptibility to develop endometriosis. The choice of which polymorphisms were to be analysed fell to WNT4, VEZT and FSHB genes because of their hypothetical correlation with endometriosis, according to recent findings in the literature. Several GWAS meta-analyses have reported a possible pathogenetic role [16,27]. In particular, an association study published in 2017 analysed these SNPs in a Greek population, highlighting a significant connection with the disease [29]. The molecular biological examinations of single-substitution polymorphisms in our study were carried out from whole blood drawn exclusively from Sardinian patients, upon their informed consent. We chose to limit the study to only patients of Sardinian origin, considering the peculiarity of this island, which can be seen as a genetic macro-isolate. The choice of limiting the selection of the target population to patients of Sardinian origin for at least three generations was not intended to exclude the possible influence of genetic features from non-Sardinian distant ancestors. It was, rather, an attempt to make the target more homogeneous.

The genetic analysis of complex traits is simplified in isolated populations such as this one, in which inbreeding and the “founder effect” reduce the genetic diversity of complex and polygenic diseases such as endometriosis. The goal was to identify a genetic characterization of the disease in the Sardinian population, to shed light on the etiopathogenetic mechanisms of endometriosis, and to provide predictive markers for a non-invasive diagnosis of the disease.

#### Patients and Study Design

The present clinical study represents the initial application, on a small scale, of a study protocol conducted at the Department of Obstetrics and Gynaecology of the Hospital “Duilio Casula” of Monserrato, University of Cagliari, in collaboration with the Laboratory of Genetics and Genomics of the Pediatric Hospital Microcitemico “A. Cao” of Cagliari. Written consent was obtained from the local ethics committee (EndoSNPs, Prot. PG/2019/13157). In line with the Declaration of Helsinki 1975, revised in Hong Kong in 1989, the clinical trial was registered (ClinicalTrials.gov ID: NCT02388854). This study involved a total of 72 women who underwent surgery, 41 representing the cases with clinical and histological diagnosis of endometriosis, and 31 representing the corresponding controls—women without a diagnosis of endometriosis who underwent surgery for benign indications different from endometriosis and chronic pelvic pain. For each patient, a data collection folder was prepared in a database elaborated for the purpose. Apart from personal records, data on geographic origin and anthropometric measurements (weight, height, body mass index (BMI)), the data collection form contained data on habits such as cigarette smoking and alcohol intake, and clinical data on possible co-morbidities or previous surgical interventions. In patients with a diagnosis of endometriosis, data on this pathology were obviously included (familiarity, staging, location of lesions, transvaginal ultrasound data, gynaecological examination, symptomatology, number and type of surgical operations, histological data), type and duration of drug therapy (nonsteroidal anti-inflammatory drugs (NSAIDs), hormone therapy and antibiotics). The diagnosis of endometriosis was clinical, related to ultrasound and histology reports. The staging used was the revised classification proposed by the American Fertility Society/American Society for Reproductive Medicine (AFS/ASRM) [30], which is still the most widespread and widely used, in clinical and academic fields, to describe the severity of endometriosis in a standardized format. Localizations of deep infiltrating endometriosis (DIE) infiltrating the bladder or bowel were classified as severe. For each main symptom (dysmenorrhea, chronic pelvic pain, dyspareunia, intestinal symptoms, urinary symptoms), a score was assigned based on the intensity perceived by the patient (visual analogue scale (VAS) 0–3: mild; 4–7: moderate; 8–10: severe). For the controls, women who had never had a diagnosis or a suspicion of endometriosis were selected. The close and remote pathological medical history of these patients was also negative for dysmenorrhea, dyspareunia, chronic pelvic pain and infertility/sub-fertility, as one of the inclusion criteria was to have given birth to at least 2 children. A common criterion to be included in the study was geographic origin. All participants had an exclusively Sardinian origin for at least 3 generations (maternal and paternal grandparents from Sardinia).

## 2. Results

### 2.1. Review of the Literature

#### 2.1.1. Gene Polymorphisms

Single nucleotide polymorphisms are present in the population with an allelic frequency >1%, and represent the most common type of genetic variation in humans (90% of polymorphisms). They are present with alternative allelic variants and follow a Mendelian inheritance. Their wide dissemination in the genetic makeup (one in every 100–300 base pairs) makes them interesting in studies on allelic association and usable as molecular markers. Single nucleotide polymorphisms (SNPs) can generate synonymous or non-synonymous mutations if present in coding regions of genes, or lead to alterations of the gene product if present in intronic or intergenic regions [2]. Variations in the genome may determine different individual susceptibilities to a disease, or different pharmacological responses. SNPs are therefore useful to identify “disease genes” through association studies based on population screening [2]. Technological advances in the field of molecular biology, especially through polymerase chain reaction (PCR) techniques, have stimulated interest in identifying certain gene polymorphisms and their influence on susceptibility in developing endometriosis. The choice of candidate genes falls to those genes whose polymorphic variants are hypothesized to be involved in the pathophysiological molecular mechanisms underlying the disease: genes involved in steroidogenesis and in receptorial activity of sex hormones, genes involved in inflammatory and immune response processes, genes regulating metabolism, genes affecting the processes of tissue remodelling and neoangiogenesis, and DNA repair genes [31,32].

#### 2.1.2. Genes Involved in Stereidogenesis and Receptorial Activity of Sex Hormones

Endometriosic and endometrial tissues are responsive to the effect of sexual steroid hormones, especially oestrogens. It is no coincidence that the risk factors of this disease include early menarche and late-onset menopause, conditions that notoriously involve extended exposure to endogenous oestrogens throughout life. Therefore, among the most common pathogenetic hypotheses is that there is possible dysregulation of the ligand-receptorial signalling involving the main sex hormones, oestrogens and progesterone [33,34]. The choice of candidate genes for various association studies is based on this mechanism. The two estrogenic receptor (ER) isoforms (ER-beta and ER-alpha) are codified by two genes (ESR2 and ESR1) with tissue-specific distributions, and have different capacities in connecting ligands (oestrogen and antioestrogen) and starting the transcription of target genes. The influence of ESR1 gene polymorphisms was studied in women affected by endometriosis in different populations, both European and Asian, but with inconsistent results. For the ESR2 gene, the studied polymorphism is localized in region 3′UTR of the gene, on nucleotide 1730 (1730 adenine (A) → guanine (G)), and it is recognized by the restriction enzyme AluI. This polymorphism has been related to a risk of severe stage IV endometriosis in women of Japanese origin [35], but no significant association with the development of the disease in Italian and Korean women was seen [36]. The pathogenetic hypothesis that there is involvement of the corresponding receptors and their malfunctioning in determining those phenomena involved in the development of endometriosis is based on the dysregulation of tissue sensitivity influenced by progesterone. In particular, as an object of numerous studies on their role in the onset and development of ovarian carcinoma, the polymorphism of progesterone receptor (PR) gene, called PROGINS, seems to compromise the perfect receptor–ligand functionality in target tissues, making them less sensitive to progesterone and altering the hormonal balance in favour of oestrogenic activity. The PROGINS polymorphism was found more frequently in women suffering from endometriosis. A statistically significant association was found in affected women from Italy [37], Austria [38] and Brazil [39], while similar studies on women from India [40], Australia [41] and Germany [42] came to exactly opposite conclusions. Other more or less recent studies focused their research on gene polymorphisms of the androgen receptor (AR). Androgens play an important role, although not sufficiently clarified, in endometrial physiology and physiopathology, counteracting the effect of oestrogen on cell proliferation [43,44,45,46]. The AR gene is localized on the Xq11.2-q12 chromosome; the corresponding protein has eight domains with transactivation function, all distributed in exon 1. Here, we find a polymorphic microsatellite consisting of a cytosine–adenine–guanine (CAG) sequence, variously repeated in individuals, on which the length of glutamine residues in the amino-terminal end of the AR protein depends, with repercussions on its functionality [47,48]. In this case, the results are very different regarding the ethnic group used as a sample. Shaik’s work on a female Indian population concluded that the presence of 19–21 CAG repeats of the AR gene could be considered a susceptibility marker for endometriosis and uterine leiomyoma [49], confirming the results obtained from a previous work on Taiwanese women [50], but in stark contrast to Lattuada’s results on Italian women [51].

#### 2.1.3. Genes Involved in Inflammation and Immune Response

The development of the inflammatory process is an integral part of the pathophysiology of endometriosis. Polymorphisms of numerous cytokines have become the objects of many studies, but with conflicting results. In their work on a female Taiwanese population, Hsieh et al. demonstrated an association between susceptibility to endometriosis and the presence of gene polymorphisms: –509C/T in the region promoter of the transforming growth factor-β1 (TGF-β1) gene, –627A/C in the region promoter of the interleukin 10 (IL-10) gene, and 881T/C in the IL2 beta receptor gene [52]. A 2017 study focused on TGF-β1, in particular on how it affects the development and progression of the disease with regard to hypoxia. The results support the hypothesis that TGF-β1 is involved in the pathogenesis of endometriosis through the regulation of vascular endothelial growth factor (VEGF) expression. TGF-β not only increased its levels, but also had an additive effect at the transcriptional level on hypoxia itself, increasing the expression of VEGF up to 87% [53]. The polymorphisms of the gene coding for cytokine TNF-α, with regard to its key role in the development of the acute phase of inflammatory processes and regulation of the immune response, have also been at the centre of several works: variations in promoter regions do not appear to be associated with an increased risk of endometriosis in Korean, Taiwanese and Caucasian women [54]. Several lines of evidence prove that endometriosis is characterized by specific modifications of the immune system, in particular endowing endometriosic cells with the ability to elude the mechanisms of immunosurveillance and thus guarantee easier implantation and development in ectopic sites. Among these mechanisms, we can hypothesize changes in the expression of human leukocyte antigens (HLAs), important for immunological recognition, secretion of circulating antigens competing with surface antigens, critical for self/non-self-recognition, and production of immunosuppressive or proapoptotic factors against T cell effector [55]. The HLA system or major histocompatibility complex (MHC) is the gene complex which boasts the largest number of polymorphisms in the human genome. Wang’s study of 2001 on Chinese women reported the influence of polymorphisms of the HLA-B gene, but not the HLA-A gene, on the susceptibility to the development of endometriosis [56], whereas Ishi in 2003 reported a significant association with HLA-DRB1*1403 and HLA-DQB1*03031 alleles in Japanese women, but not with HLADPB1 [57]. In this case, too, we detect conflicting results. PTPN22 is a coding gene for a protein involved in the responsiveness of B and T lymphocyte receptors, whose mutations may be responsible for the onset of autoimmune diseases. The associations observed between PTPN22 (C1858T) and the risk of endometriosis suggest that this polymorphism could be a useful marker of susceptibility to the disease [58].

#### 2.1.4. Genes Involved in the Processes of Tissue Remodeling and Neoangiogenesis

Vascular endothelial growth factor (VEGF) and endothelial growth factor receptor (EGFR) are molecular factors known to be involved in the regulation of neoangiogenesis, tissue remodelling and proliferation, phenomena also necessary for the growth of ectopic foci of endometrial tissue. Specifically, VEGF induces the proliferation, migration and differentiation of endothelial cells and capillary formation. Several studies have shown high levels of VEGF in peritoneal fluid and serum, and increased expression of mRNA and proteins in patients with endometriosis [59]. Furthermore, VEGF causes an increase in vascular permeability and a release of proteases such as metalloproteases (MMPs), enzymes able to “cut” the proteins of matrix and basement membrane, important for cellular invasion and tissue remodelling. It can also prevent apoptosis of different cell types. Perini’s group study in 2014 on Brazilian women indicated a risk association with –1154G> polymorphism A of the VEGF gene [59]. EGFR is a transmembrane glycoprotein which plays an important role in the control of growth, differentiation and cell motility. The EGFR +2073A/T polymorphism, a candidate as a susceptibility gene, was studied by Hsieh et al. in 122 Taiwanese women and 139 controls, showing an association with a high risk of disease. These results were subsequently denied when reported to the Japanese population [60].

#### 2.1.5. Genes Regulating Metabolism

Glutathione S-transferase mu M1 (GSTM1), glutathione S-transferase P1 (GSTP1), glutathione S-transferase theta T1 (GSTT1), N-acetyltransferase 1 (NAT1) and N-acetyltransferase 2 (NAT2) genes all encode phase II enzymes involved in the metabolism of xenobiotics, including toxic compounds such as dioxin and polycyclic aromatic hydrocarbons. In particular, a 2015 meta-analysis by Guo on the association between variants of GST, GSTM1 and GSTT1 genes, which encode for glutathione S transferase, and the development of endometriosis showed a significant increase in risk. Women with GSTM1-null genotype compared to other genotypes recorded a summary OR of 1.96, demonstrating an approximately double risk of endometriosis. Women with GSTT1-null genotype compared to other genotypes recorded a summary OR of 1.77, showing an 80% risk of endometriosis. Nevertheless, the existence of bias in this meta-analysis indicates that the magnitude of risk could be smaller or even non-existent [61]. Many gene polymorphisms corresponding to phase I enzymes involved in, among other things, oestrogen metabolism were investigated as variants associated with a greater susceptibility to disease, for example CYP1A1 6235T/C, CYP17A1 –34T/C and CYP19A1 microsatellite TTTA repeat [62,63].

#### 2.1.6. Genes Related to the Process of DNA Repair

Many studies have proposed oxidative stress as a factor involved in the pathophysiology of endometriosis [64]. An excess of reactive oxygen species causes DNA damage, base modifications and chromosomal aberrations, which would justify the metaplastic features of the disease. The DNA repair system from oxidative stress involves the intervention of enzymes encoded by genes including X-ray repair cross-complementing 1 and 3 (XRCC1 and XRCC3) and XPD (also known as excision repair cross-complementing (ERCC)), whose defective functioning decisively contributes to develop endometriosic lesions. Attar et al. in 2010 reported no significant differences between the frequency of genotypes and polymorphic alleles of APE1, XRCC1, XPD, XPG and HOGG1 genes in patients with and without endometriosis, whilst a significant increase in disease frequency would appear to be associated with the XRCC3 Thr/Thr genotype [65].

#### 2.1.7. Genome-Wide Association Study (GWAS) and Next-Generation Sequencing (NGS)

So far, we have discussed works adopting an investigative approach based on the a priori selection of polymorphisms of candidate genes known to have hypothetical biological roles in the pathogenesis of endometriosis, and their subsequent research on DNA samples. A radical change in perspective was applied in the so-called hypothesis-free methods, such as the analysis of family linkage, association studies on the genomic scale, or genome-wide studies (GWASs), up to the most recent new generation sequencing technique, next-generation sequencing (NGS), which targets the entire genome, without pre-selecting a particular gene or gene region, therefore with no initial pathogenetic hypothesis. Among its main advantages, this type of approach can detect alterations of unexpected molecular pathways perhaps shared by apparently very different pathologies. Analyses of family linkage and genome-wide association studies are different though complementary approaches to the identification of genetic risk variants across the whole spectrum of allele frequencies. A linkage analysis aims to identify those genetic variants which are rare in the general population, but are responsible for the family aggregation of a given pathology, while a GWAS aims to identify genetic common variants in the general population associated with the risk of the disease [66]. Studies of family linkage have analysed the association between a known marker and the unknown gene/disease, assuming their proximity to the chromosome and the consequent co-segregation in affected families. In 2005, an impressive linkage study was carried out by Treloar’s group on 1176 English and Australian families, each with at least two members, mainly sister pairs, with a surgical diagnosis of endometriosis, for a total of 4985 genotyped individuals, 2709 of them women with endometriosis. This study found a strongly significant linkage on chromosome 10q26 and another region suggestive of linkage on chromosome 20p13 [67]. The fine-mapping of the 10q26 region using 11,984 SNPs in 1144 affected families and 1190 controls was described in a study by Painter’s group in 2011, which identified three signs of significant association with endometriosis [68]. Among them, the only one to be replicated in an independent cohort was rs11592737, located in the CYP2C19 gene, an important candidate gene involved in oestrogenic metabolism, particularly in the conversion of oestradiol to oestrone [69]. However, despite the valuable contribution of this method to the study of monogenic pathologies, this technique was shown to be less accurate when applied to multifactorial diseases such as endometriosis, able to identify extensive chromosome regions containing hundreds of genes with too large and inaccurate linkage peaks. This fact, together with the lack of sensitivity and the difficulty of independently replicating the positive results related to the need for a large number of affected families with more generations available to test, has made this method obsolete, opening the way to the more promising GWAS. A GWAS is a survey on all, or almost all, genes of an individual of a particular species to determine the gene variations between them. Later, attempts are made to associate the observed differences with some specific traits, such as a disease. In these cases, samples from hundreds or thousands of individuals are evaluated, usually looking for single nucleotide polymorphisms (SNPs). Instead of reading whole gene sequences, these systems usually identify SNP markers of groups of gene variations (haplotypes). If some gene variations are significantly more frequent in sick individuals, then the variations are said to be associated with the disease. These variations are then considered indicative of the region where the disease-causing mutation is likely to be found. The first GWAS on endometriosis was carried out on a Japanese population in 2010, on a sample of 1432 cases and 1318 controls. The study identified a significant association (*p* = 5.6 × 10^−12^; OR 1.44 (1.30–1.59)) with the SNP rs10965235 located on the CDKN2B-AS1 gene on chromosome 9 and with the SNP rs16826658 (*p* = 1.7 × 10^−6^ OR 1.2 (1.11–1.29)) located on the WNT4 gene on chromosome 1 [18]. The first gene regulates some onco-suppressors such as CDKN2B, CDKN2A and ARF; its inactivation has been correlated with the development of endometriosic foci and endometrial carcinoma [70]. The second one is a very important gene involved in the development of the female genital apparatus, indispensable for the formation of Müllerian ducts [12]. It has a sequence that regulates ESR1 and ESR2 genes, and it is still among the main candidate genes for endometriosis and ovarian cancer. A subsequent GWAS of 2016 also focused on this gene. Using a sample of 7090 individuals (2594 cases and 4496 controls), the study found the marker in the region of the WNT4 gene, with the strongest association with the risk of endometriosis: the SNP rs3820282 [71]. In 2011, a subsequent GWAS was conducted through the International Endogene Consortium (IEC) by Painter’s group on British and Australian women, analysing 3194 cases of surgically diagnosed endometriosis and 7060 controls [19]. The study divided the sample of affected individuals into two categories based on the severity of the pathology (stage I–II and stage III–IV), and detected a strongly significant linkage, in particular in the “severe” subgroup, with two SNPs: rs1250248 (*p* = 3.2 × 10^−8^; OR 1.30 (1.19–1.43)), located on the FN1 gene on chromosome 2, involved in cell adhesion and migration, and rs12700667, (*p* = 1.5 × 10^−9^; OR 1.38 (1.24–1.53)), located on an intergenic region of chromosome 7p15 containing regulatory elements, located upstream of three candidate genes: miRNA148a (88kb), the closest, encoding a microRNA implicated in the important Wnt β-catenin signalling pathway for the homeostasis of sex hormones; NFE2L3 (331 kb), encoding a transcription factor involved in the processes of phlogosis, differentiation and carcinogenesis, highly expressed in the placenta; and the HOXA 10-11 genes, coding for homeobox A transcription factors, considered important in the spatial and temporal regulation of uterine development about 1.35 Mb downstream. Thanks to the data of this GWAS, it was also possible to refute the conclusions of later works, such as the study of Grechukhina in 2012, which reported a high frequency (31% vs. 5% case-control) of SNP rs61764370 in the 3’UTR region of the Kirsten rat sarcoma viral oncogene homolog (KRAS) gene in women with endometriosis [72]. The KRAS gene was considered a strong candidate gene; in fact, KRAS activation determined the development of peritoneal endometriosis in 47% of mice and endometriosis-like ovarian lesions in 100%. SNP rs61764370 interrupts the link between the LCS6 complementary site and let-7 miRNA, altering the expression of the KRAS gene and favouring the development of the disease. The analysis conducted in 2012 by Hien Luong and Painter denied this risk association [73]. A meta-analysis of the GWASs of the IEC and the Japanese group, carried out in 2012, confirmed the associations previously reported in the respective studies and identified further significant new loci of association localized on GREB1 genes, involved in estrogenic regulation; VEZT, involved in cell growth, migration and adhesion; and ID4, an ovarian tumour suppressor [17]. Another interesting GWAS of 2017 was a meta-analysis of 11 case-control datasets, for a total of 17,045 cases of endometriosis and 191,596 controls. Five additional new loci significantly associated with the risk of endometriosis (*p* < 5 × 10^−8^) of genes involved in sexual steroid hormone pathways (FN1, CCDC170, ESR1, SYNE1 and FSHB) were identified, for a total of 16 genomic regions associated with endometriosis risk in one or more populations [27]. At present, 19 independent single nucleotide polymorphisms (SNPs) have been validly associated with endometriosis, explaining 5.19% of disease variance [27]. In particular, a recent study investigated in a Greek population three of these SNPs: rs7521902, rs10859871 and rs11031006, mapped respectively on WNT4, VEZT and FSHB genes, highlighting a significant association with endometriosis [29]. In recent years, thanks above all to the reduction in cost and time of examination, an innovative DNA reading technique has come forward forcefully: next-generation sequencing (NGS). Through NGS, it is possible to analyse a large number of DNA fragments in parallel, until the sequences of many genes are obtained simultaneously, even the entire coding region (exome sequencing) or the whole genome (whole genome sequencing) of an individual. Instead of screening several variants, the most common ones, scattered throughout the genome as in the genome arrays of GWASs, with an NGS reading it is possible to search for the rarest variants in a population. A study of families affected by endometriosis using NGS was conducted in 2014 by Chettier et al.; out of 13 pairs of affected sisters, five shared genomic regions (in chromosomes 2, 5, 9 and 11) where rare gene variants associated with endometriosis are located [74]. In February 2016, Er’s research group focused on the molecular characterization of endometriosis associated with ovarian cancer through targeted NGS of 409 cancer-related genes by identifying gene variants on the PIK3CA and ARID1A genes, related to endometriosis with a high risk of malignant transformation [75]. Most GWA variants associated with the risk of endometriosis are not codified and the genes responsible for association signals have not been identified. A direct approach to identifying target genes could be identifying variants encoding common proteins associated with the risk of endometriosis. Large international exon sequencing projects have identified exome variants modifying the coding of corresponding proteins (nonsynonymous codification, alternative splicing sites or stopping codon gain or loss), classifying them according to their frequency. The role of protein-coding variants associated with endometriosis risk was therefore directly tested by genotyping through a specific Illumina array (Illumina exome chip) designed to allow the simultaneous capture of rare, low-frequency and common frequency exonerated variants. In all, 7164 patients with endometriosis and 210,005 controls were analysed. The initial association between disease and variations in encoding the CUBN, CIITA and PARP4 genes was not confirmed when replicating the study [76]. There is therefore no clear evidence of the existence of codifying variants of common or low-frequency proteins which can influence the risk of endometriosis. Variants capable of directly modifying the sequence and function of protein amino acids would have provided immediate gene targets. The causal SNPs are probably located in non-codifying sequences, but are involved in functions regulating gene expression and are able to influence the development and progression of disease [77]. If GWASs have identified various susceptibility loci, the biggest challenge of the post-GWAS era is therefore to understand the functional consequences of these loci. Studying the association between genetic variations and gene expression offers a way to link risk variants with the corresponding target genes. Gene expression varies in different individuals, and the level of expression of certain genes is under the control of particular regulatory variants indicated as expression quantitative traits loci (eQTLs), which can influence the abundance of specific transcripts [78]. Various study groups have identified three eQTLs capable of altering the transcription of potential target genes: LINC0039 and CDC42 on chromosome 1, CDKN2A-AS1 on chromosome 9 and VEZT on chromosome 12. Further functional studies are needed to confirm the role of causal genes in different susceptibility loci [79].

#### 2.1.8. Epigenetics

During the past five years, evidence has emerged that endometriosis may be an epigenetic disease. Epigenetics proves to be the common denominator of a pathology in which both genetic and environmental components play pathogenetic roles. Many large-scale profiling studies on gene expression have unequivocally demonstrated that many genes are deregulated in endometriosis [80]. Several aspects related to possible post-transcriptional interference that may result in changes in endometriosic tissues, compared to healthy ones, have been investigated. These modifications are expressed at different levels (transcriptional, post-transcriptional, post-translational) and through different mechanisms (methylation, histone modification, activators, repressors, miRNAs). In particular, changes in terms of hypermethylation and silencing of genes which are normally expressed during the proliferative phase of the cycle and regulate the modalities of cell adhesion have been studied, such as HOXA-10, E-cadherin, the GATA family of transcription factors and the group of nuclear receptors for steroids (SR), which includes the progesterone receptors in two isoforms, PR-A and PR-B. The opposite transcriptional activity of the two isoforms has been demonstrated to be due to binding of the Hic-5 differential co-regulator. This type of change affects the resistance of endometriosis to treatment with progesterone [81]. Expression by endometrial tissue of the two receptors for progesterone, PR-A and PR-B, and their relationship influences the reaction of tissue to exposure to progestins. A prevalence of PR-A results in resistance to therapy and loss of cell adhesion. In vitro studies have shown that deacetylase inhibitors show promise as therapies for the treatment of endometriosis. In particular, trichostatin A has been shown to reactivate the expression of E-cadherin, an intercellular adhesion protein, which is hypermethylated and therefore silenced in endometriotic cell lines, with a concomitant reduction in invasiveness [80]. MiRNAs are a class of endogenous NcRNAs of about 22 nucleotides, which play a key role in regulating gene expression. As well as DNA methylation and histone modification, they implement this regulation without any change in DNA sequence, but inhibit or induce miRNA degradation. A profiling study of miRNA expression identified 48 out of 287 miRNAs differentially expressed in healthy women and women with endometriosis. The target genes of these miRNAs include many genes involved in endometriosis (Erα, ERβ, PR, TGFβ), suggesting that miRNA deregulation may play a pathogenetic role [80]. Acetylation, methylation and phosphorylation are the most well-known and studied post-translational histone modifications. These changes alter the interaction of histone with DNA and nuclear proteins. In general, histone acetylation is associated with transcriptional activation and deacetylation with transcriptional repression [82,83]. Studies on animals seemed to suggest that HDAC2 is expressed aberrantly in ectopic endometrium. Furthermore, Sun et al. (2003) reported that treatment of endometriotic stromal cells with prostaglandin E2 (PGE2) resulted in increased acetylation of histone H3 bound to StAR promoter, with concomitant induction of co-activator CBP/p300, binding of CREB protein, and possible involvement in the pathogenesis of endometriosis [84]. So far, studies on epigenetics have barely scratched the surface of this area, and further research will allow us to shed light on the pathophysiology of endometriosis, and on the complex mechanisms regulating the activation or silencing of different genetic targets. Such knowledge will provide a basis for the identification of new drugs and diagnostic biomarkers.

### 2.2. Cohort Study

The patients’ average age was 35 years (median 33 years in the group of cases, 36 years in the group of controls). The median values of body mass index (BMI) in the two groups were uniform (21 for cases vs. 22 for controls). Among the affected women, we found comorbidities in nine: two cases of haemorrhoidal disease, one of serous cystadenoma, borderline mucinous cystadenoma associated with breast cancer, mitral valve prolapse, umbilical hernia, anxiety disorder, hyperinsulinemia, and irritable colon. Among the controls, nine women had a pathological medical history: HashimotO’s thyroiditis (5), type 2 diabetes mellitus, hypercholesterolaemia, uterine prolapse (1), renal ptosis (1), gastro-oesophageal reflux disease (1), breast fibroadenoma (1) (Table 1).

In the sample of women diagnosed with endometriosis, six were at stage I (minimum disease), with superficial lesions; five were at stage II (mild disease), with extended foci (up to 4 cm) in different locations; 14 were at stage III (moderate disease), mostly cases of ovarian endometrioma; and 16 were at stage IV (severe disease), in which we included cases of deep endometriosis with extended nodular formations (5–6 cm) and infiltrating contiguous structures such as the rectum, pararectal spaces, ureters and retroperitoneum. Table 2 shows, for each stage, the data related to age and BMI at the time of diagnosis, which was almost uniform among the subgroups; number of term pregnancies; surgical interventions for the treatment of the disease, performed laparoscopically in all cases, two completed with intestinal resection; and extent and severity of symptoms (the percentage of symptoms perceived as severe (score 3) was higher in stage IV (94%) than in stage I (40%)). In stages II and III, the degree of symptom severity perceived by women showed similar scores (60% vs. 64%), proving that symptomatology is not always connected linearly to the surgical stage of the disease. None of the women in the group of affected patients reported a family medical history positive for endometriosis.

The frequency of genotypes observed in the groups of the reference population, 72 individuals, was globally consistent with the Hardy–Weinberg equilibrium law (Table 3).

Applying the HWE test to all components of the study population (cases + controls), there are no consistent differences between expected and observed genotypic frequencies, as shown by the *p*-values calculated for each investigated SNP. The *p*-values obtained are equal to 0.2396 for rs11031006 (FSHB), 0.5766 for rs10859871 (VEZT) and 1 for rs7521902 (WNT4); they allow us to accept the starting hypothesis, according to which there is no difference among the examined groups concerning the considered parameters. For each SNP, checking the expected and observed genotypic/allelic frequencies in single groups of affected cases and healthy controls, a modest significance in deviation from the Hardy–Weinberg equilibrium is highlighted, regarding the group of positive controls for genotypes containing allele C of the polymorphism rs10859871 (VEZT) (HWE *p*-value = 0.0435.) This violation of the proportions of the Hardy–Weinberg equilibrium could constitute a bias selection (Table 4).

The following data are related to the study on genetic association between the considered SNPs and endometriosis.

rs11031006 (FSHB): Patients with endometriosis did not have an increase in genotypic or allelic frequency for this polymorphism compared to the control group (*p* = 0.9999, OR = 0.000, 95% CI, 0.000–15.000 and *p* = 0.731, OR = 1639, 95% CI, 0.393–683, respectively, for heterozygous genotype GA and polymorphic minor allele A) (Table 5, Figure 1). We also verified whether, by stratifying the sample of cases based on the severity of the situation (considering only women with stage III and IV endometriosis), a possible association with the investigated polymorphism would emerge; however, no correlation was found.

rs10859871 (VEZT): In the case of the polymorphism of the VEZT gene, we found a significant difference in frequency of homozygous genotype CC in the healthy control group compared to the affected women (*p* = 0.0111, OR = 0.0602, 95% CI, 0.005–0.501) (Table 6).

The same result was replicated by applying the analysis to the sub-group of affected women with moderate to severe endometriosis (stage III–IV) versus healthy controls, confirming a protective role for the homozygosity of the allelic variant (*p* = 0.02703). No significant differences between the groups concerning the allelic frequency of the minor allele C (*p* = 0.2627, OR = 0.6248, 95% CI, 0.300–129) were reported (Figure 2).

rs7521902 (WNT4): Patients with endometriosis did not have an increase in genotypic or allelic frequency of this polymorphism compared to the control group (*p* = 0.3088, OR = 0.4133, 95% CI, 0.10–1.8 and *p* = 0.3297, OR = 2257, 95% CI, 0.55–914, respectively, for heterozygous genotype CA and polymorphic minor allele A) (Table 7, Figure 3). No association was found between this SNP and the disease, even when limiting the analysis to the sample of severe surgical cases (stage III and IV).

## 3. Discussion

Our analysis of the recent literature shows evidence of many genes possibly implicated in different pathogenetic mechanisms of endometriosis (Figure 4).

Nevertheless, it is clear that there are differences in genetic associations in endometriosis between different populations around the world. Therefore, it is important to study the genetic basis of this condition in several populations and to replicate the previous results, in order to define the role of significant variants of the risk of endometriosis. In this study, we specifically wanted to test the association between rs11031006 (FSHB), rs10859871 (VEZT) and rs7521902 (gene polymorphisms) and greater susceptibility to the development of endometriosis in a native Sardinian population. SNP rs7521902 WNT4 has been repeatedly associated in the literature with greater susceptibility to endometriosis in women of different ethnicities, including British, Australian, Italian and Japanese [10,16,17]. In the present study, no association between the risk C allele of this SNP and increased risk of endometriosis was detected. These data are consistent with those previously reported by Sundqvist et al. and Mafra et al. [9,13] in a Belgian population. Similarly, no association was detected in a study conducted on 800 Brazilian women (400 with and 400 without endometriosis) [9], or in another conducted on Chinese women (646 with and 766 without endometriosis) [14]. The SNP rs11031006 (FSHB) is another polymorphism candidate for a pathogenetic role, under its possible effect on the inhibition of hormonal feedback; although a significant association in a Greek population was reported, our results in a Sardinian population do not appear to be in line with the literature. The SNP rs10859871 is located in chromosomal region 12q22, 17 kb upstream of the VEZT 187 gene. VEZT is a transmembrane protein with a short extracellular and long intracellular domain, which clips on myosin VIIA as part of the adherent junctional complex in epithelial cells [21,22]. In the blood and endometrium, the C allele at risk of endometriosis of rs10859871 SNP in previous works was associated with increased expression of VEZT, and a pathogenetic role had been hypothetically attributed to this over-expression of VEZT [20]. In the case of gene polymorphism rs10859871 (VEZT), in our study, we instead show a significant and unexpected protective role in the pathology, in sharp contrast to the results obtained in other association studies conducted on populations with different ethnicities. We believe that this is the result of bias due to the small sample. As proof, the preliminary calculation of the Hardy–Weinberg equilibrium indeed signalled a significant deviation precisely in the group of positive controls for this polymorphism, compromising the validity of the results. Therefore, the obtained results did not highlight a significant association between the studied gene variants and a greater risk of developing the disease, although several other studies in the literature show the opposite. This is not surprising, if we consider that the obtained results refer to a Sardinian population known for its unique characteristics. The Sardinian population has always aroused considerable interest in human genetics, both for the peculiar distribution of different genetic variants and for the numerous genetically based diseases particularly frequent on the island. The Sardinian population, even if placed within European variability, shows singular characteristics: some frequent variants in Sardinia are rare or even absent in other populations, and vice versa. For example, the lack of significance of the SNP rs7521902 WNT4 as an endometriosis risk variant in the population of our study is largely due to a minimum allelic frequency (MAF), very low in the Sardinian population (0.0971). In other words, if a gene variant in a given ethnic population is quite rare, a large sample of individuals will be needed to test its frequency in terms relevant to the study. Similarly, the results in our study related to the SNP rs10859871 (VEZT), curiously not in line with what is highlighted in the literature, may have experienced interference because of the reduced size of the sample. It is equally true that most of the SNPs analysed in association studies or detected in GWASs do not fall within regions codifying the gene, and functional studies on them are insufficient. It is easy to hypothesize that the regulatory effect of these polymorphic variants on the expression of certain genes is in turn influenced by the modulation of other factors and co-factors expressed on other regions of the genome whose location and function we do not know. There is also more and more evidence for the role of epigenetic influences, which, through environmental, food-related and/or misunderstood factors and through methylation mechanisms of sequences and histone acetylation, can induce modifications in DNA gene expression without changing its sequence, but simply by turning on or off certain target genes. Since the pathogenesis of endometriosis is extremely complex, involving both genetic background and environmental conditions, studies with conflicting results in several cases have made it difficult to interpret these data. The lack of confirmation of the previous results is therefore largely attributed to an insufficient sample, to population differences and the interaction with genetic and/or non-genetic factors. For this reason, it appears mandatory to extend the case histories of the study in order to reveal or confirm the influence of these gene variants on the Sardinian population.

## 4. Materials and Methods

### 4.1. Sampling and Genotyping

After collecting their informed consent, we took a sample of whole blood from each patient (9mL EDTA tube) for genetic analysis. On each sample, a label indicating the date of acquisition and an alphanumeric code (single coding) was affixed, through which only authorized personnel could trace the patient’s identity through a special data collection file. The samples were stored in a freezer at a temperature of −30 °C, locked and accessible only by authorized personnel. Genomic DNA was extracted from whole blood with a Qiagen Extraction Kit (QIAmp ^®^ DNA Mini Kit (QIAGEN GmbH, Hilden, Germany), according to the protocol illustrated in the kit manual. Subsequently, genomic DNA extracted from whole blood was read in a NanoDrop photometer 3300^Tm^ ThermoFisher Scientific (www.termofisher.com) at wavelengths of 260, 280 and 260nm/280nm, to assess its quality and quantity. The extracted nuclear DNA was subsequently aliquoted for the next phases of PCR. The analysis of rs7521902, rs10859871 and rs11031006 SNPs examined in the study was carried out by sequencing with the Sanger method. For each selected SNP, the minimum allelic frequency (MAF) in the Sardinian population was preliminarily verified, using the online PheWeb dataset (http://sardinia-pheweb.sph.umich.edu) (Table 8).

Each DNA sample was amplified with PCR (final volume 25 μL) according to the following protocol: to 75 ng of genomic DNA, 2.5 μL of the specific Taq Polymerase ThermoFisher Scientific (www.termofisher.com) 10× buffer and 1.5 μL of MgCl_2_ solution (25 mM) was added 0.2 μL of a 2.5 mM dNTP solution, 0.125 μL of Taq Polymerase (5 U/μL), and the forward and reverse primers (25 pmol/μLeach) specific to each analysed region. For each fragment, the same amplification file was used in which initial denaturation was foreseen at 95 °C for 10 min, 30 amplification cycles at 95 °C for 30 min, 60 °C for 30 min, 72 °C for 40 min and a final extension at 72 °C for 10 min. The PCR products were then sequenced using a BigDye Terminator v3.1 Cycle Sequencing Kit ThermoFisher Scientific (www.termofisher.com), according to the manufacturer’s instructions. The primers used for sequencing were the same ones used for the amplification of the fragments. Sequence reactions were subsequently subjected to electrophoresis with an ABI3130XL genetic analyser instrument (Applied Biosystems–ThermoFisher) and analysed with SeqScapev3.0 software (ThermoFisher).

### 4.2. Statistical Analysis

Data were tabulated in a specific database and analysed through specific software. For statistical analysis, Plink v.1.7, SPSS Statistics for Windows v.18 (IBM Corp., Armonk, NY, USA) and GraphPad Prism v.8 (GraphPad Software, La Jolla, CA, USA) software were used, applying addictive, allelic, dominant and recessive models. To examine the differences in frequency of genotypes and alleles among the groups of cases and controls, and to consider a possible association among the SNPs and the pathology under study, Fisher’s exact test was used. All of the examined SNPs had a >98% callrate. We considered a 2-tailed *p*-value < 0.05 to be statistically significant. The odds ratio (OR) and 95% confidence interval (CI) were calculated. The studied genetic variants were evaluated in advance for deviation from the Hardy–Weinberg equilibrium (HWE), comparing the observed and expected genotype frequencies in the groups of the reference sample, through Fisher’s exact test.

## 5. Conclusions

Our data underline the importance of evaluating genetic variants in different populations, in an attempt to define the genetic architecture of endometriosis and the extent of the effects of specific risk alleles in different ethnic groups. An important advantage of the analysis of genetic associations for different ethnic populations is that it can also shed light on the decomposition of the different disequilibrium linkage of populations. At the genomic level, this “trans-ethnic fine mapping” is an important method to restrict the signal to a causal variant [29]. To test this aspect further, exploration of the loci is needed, analysing the surrounding variants, for example through GWASs. GWASs allow the examination of a high number of SNPs and rare variants (up to 4.5 million) in a very large population, to find possible associations with a disease. Recent GWAS meta-analyses, identifying some genetic loci associated with endometriosis, have marked a starting point to define the molecular basis at the origin of the disease, representing the proverbial tip of the iceberg. Studies in very large populations are needed to identify new polymorphisms and to understand their molecular mechanisms. At the moment, the identified gene polymorphisms cannot be used in clinical practice to predict the risk of developing endometriosis or as diagnostic/prognostic markers, but this constitutes a concrete goal for the near future. In fact, defining the genetic alterations predisposing the onset of endometriosis has several potential benefits. First, it could allow us to identify patients at risk of developing endometriosic pathology during the pre-pubescent phase, and contribute to a non-invasive diagnosis in adulthood. Moreover, it could help to better define the molecular mechanisms responsible for its onset and, consequently, represent a rational basis for determining specific therapeutic molecules for this disease.

## Figures and Tables

**Figure 1 ijms-21-01765-f001:**
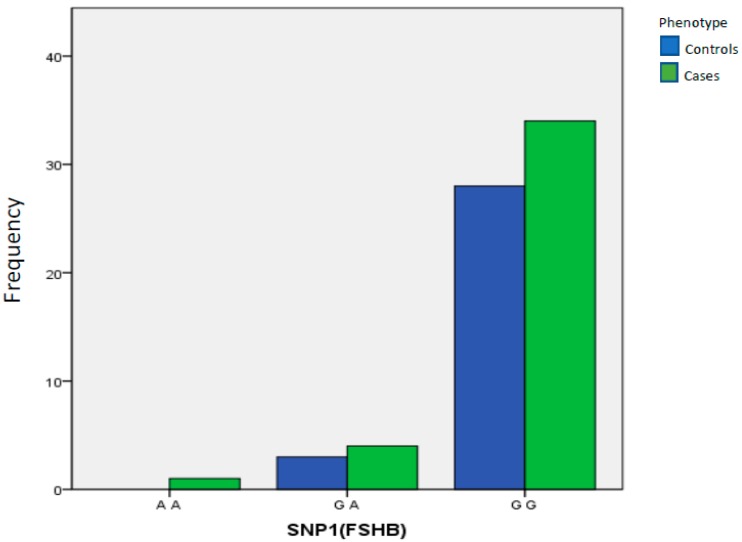
Distribution of AA, GA and GG genotypes obtained for the rs11031006 polymorphism (FSHB).

**Figure 2 ijms-21-01765-f002:**
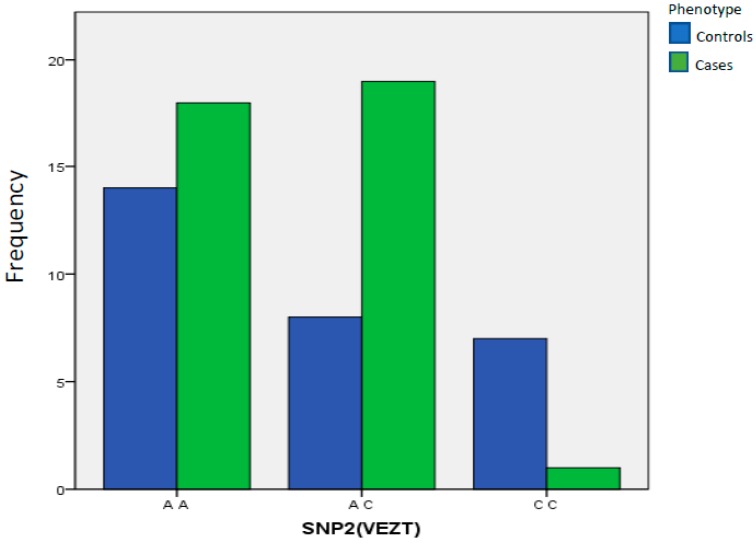
Distribution of AA, AC and CC genotypes obtained for polymorphism rs10859871 (VEZT).

**Figure 3 ijms-21-01765-f003:**
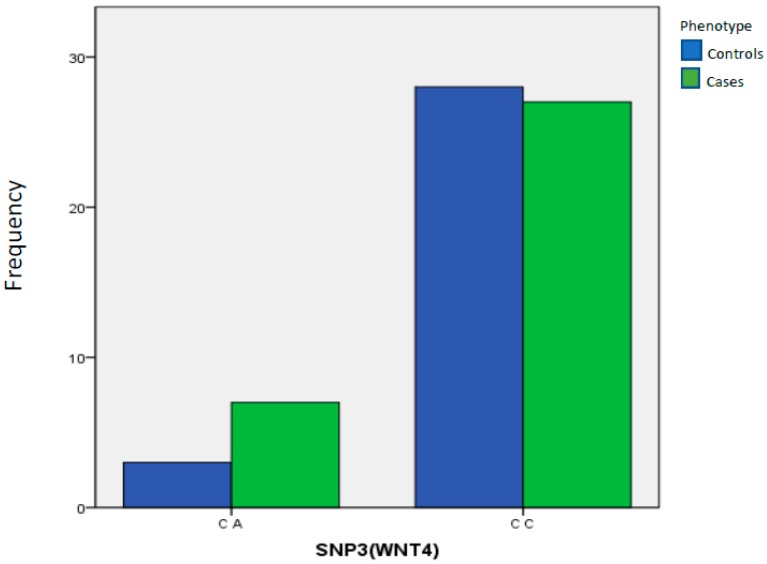
Distribution of CC and CA genotypes obtained for polymorphism rs7521902 (WNT4).

**Figure 4 ijms-21-01765-f004:**
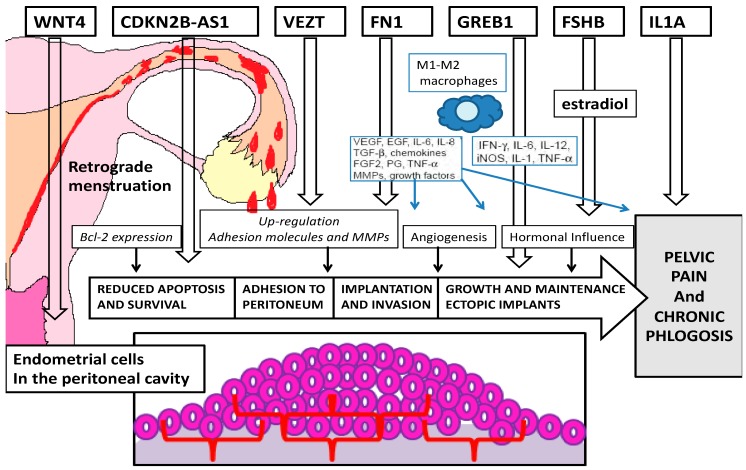
Pathogenetic targets of the most important genes related to endometriosis.

**Table 1 ijms-21-01765-t001:** Clinical and personal data of the study population. BMI, body mass index.

Clinical and Personal Data of the Study Population			
Patients	Total (n = 72)	Endometriosis (n = 41)	No Endometriosis (n = 31)
Age (median, interval)	35 (20–68)	33 (20–45)	36 (26–68)
BMI (median, interval)	21 (15–30)	21 (17–30)	22 (15–30)
Comorbidity	18	9	9
Parity		0.25	2.3

**Table 2 ijms-21-01765-t002:** Distribution of clinical anamnestic data of the group of affected women.

Stage	I	II	III	IV
**Number of patients**	6	5	14	16
**Age (median)**	35	22	37	31
**Parity (number of patients with at least one pregnancy)**	0	1	5	2
**BMI (median)**	21	20	20	20
**Number of patients with previous surgical intervention for endometriosis**	0	2	10	14
**Dysmenorrhea**				
*Mild*	2	0	0	1
*Moderate*	1	2	6	1
*Severe*	2	3	7	14
**Chronic pelvic pain**				
*Mild*	0	1	2	0
*Moderate*	2	2	3	6
*Severe*	0	0	3	6
**Dyspareunia**				
*Mild*	2	1	4	2
*Moderate*	1	2	2	6
*Severe*	0	1	3	6
**Intestinal symptoms**	2	0	7	10
**Urinary symptoms**	2	0	3	3
**Severe symptomatology percentage ***	40%	60%	64%	94%

* Percentage of women who attributed a maximum severity score (visual analogue scale (VAS) 7–10) to at least one symptom (dysmenorrhea, chronic pelvic pain, dyspareunia, intestinal or urinary symptoms); mild (0–3); moderate (4–7); severe (8–10).

**Table 3 ijms-21-01765-t003:** Expected and observed frequencies and Hardy–Weinberg equilibrium (HWE) *p*-values for SNPs investigated in total population.

SNP	TEST	A1 Minor Allele	A2 Major Allele	GENO	Observed (HET)	Expected (HET)	P
SNP1(FSHB)G>A	ALL	A	G	1/7/62	0.1	0.1203	0.2396
SNP2(VEZT)A>C	ALL	C	A	8/27/32	0.403	0.4358	0.5766
SNP3(WNT4)C>A	ALL	A	C	0/10/55	0.1538	0.142	1

**Table 4 ijms-21-01765-t004:** Expected and observed frequencies and HWE *p*-values for SNPs investigated in groups of affected cases and healthy controls.

SNP	TEST	A1 Minor Allele	A2 Major Allele	GENO	Observed (HET)	Expected (HET)	P
SNP1(FSHB)G>A	AFF	A	G	1/4/34	0.1026	0.142	0.187
SNP1(FSHB)G>A	UNAFF	A	G	0/3/28	0.09677	0.09209	1
SNP2(VEZT)A>C	AFF	C	A	1/19/18	0.5	0.3999	0.2255
SNP2(VEZT)A>C	UNAFF	C	A	7/8/14	0.2759	0.4709	0.0435
SNP3(WNT4)C>A	AFF	A	C	0/7/27	0.2059	0.1847	1
SNP3(WNT4)C>A	UNAFF	A	C	0/3/28	0.09677	0.09209	1

**Table 5 ijms-21-01765-t005:** Genotypic and allelic frequencies of rs11031006 (FSHB).

**Genotipic Test**		**Controls**		**Cases**				
		**N°**	**%**	**N°**	**%**	***P* Value**	**Odds Ratio**	**95% CI**
SNP1(FSHB)	A A	0	0.000%	1	2.56%			1.00 Reference
	G A	3	9.68%	4	10.26%	>0.9999	0.000	0.000 to 15.000
	G G	28	90.32%	34	87.18%	>0.9999	0.91	0.215 to 3.631
	Total	31	100.00%	39	100.00%			
**Allelic Test**		**Controls**		**Cases**				
		**N°**	**%**	**N°**	**%**	***P* Value**	**Odds Ratio**	**95% CI**
SNP1(FSHB)	A	3	4.84%	6	7.69%	0.7311	1639	0.393 to 683
	G	59	95.16%	72	92.30%			
	Total	62	100.00%	78	100.00%			

**Table 6 ijms-21-01765-t006:** Genotypic and allelic frequencies of rs10859871 (VEZT).

**Genotipic Test**		**Controls**		**Cases**				
		**N°**	**%**	**N°**	**%**	***P* Value**	**Odds Ratio**	**95% CI**
SNP2(VEZT)	A A	14	48.28%	18	47.37%			1.00 Reference
	A C	8	27.59%	19	50.00%	0.2931	1847.000	0.662 to 5.08
	C C	7	24.14%	1	2.63%	0.0111	0.06015	0.005 to 0.501
	Total	29	100.00%	38	100.00%			
**Allelic Test**		**Controls**		**Cases**				
		**N°**	**%**	**N°**	**%**	***P* Value**	**Odds Ratio**	**95% CI**
SNP2(VEZT)	C	22	37.93%	21	27.63%	0.2627	0.6248	0.300 to 129
	A	36	62.07%	55	72.36%			
	Total	58	100.00%	76	100.00%			

**Table 7 ijms-21-01765-t007:** Genotypic and allelic frequencies of rs7521902 (WNT4).

**Genotipic Test**		**Controls**		**Cases**				
		**N°**	**%**	**N°**	**%**	***P* Value**	**Odds Ratio**	**95% CI**
SNP3(WNT4)	C A	3	9.68%	7	20.59%			1.00 Reference
	C C	28	90.32%	27	79.41%	0.3088	0.4133	0.108 to 1.811
	Total	31	100.00%	34	100.00%			
**Allelic Test**		**Controls**		**Cases**				
		**N°**	**%**	**N°**	**%**	***P* Value**	**Odds Ratio**	**95% CI**
SNP3(WNT4)	A	3	4.84%	7	10.29%	0.3297	2257	0.557 to 914
	C	59	95.16%	61	89.71%			
	Total	62	100.00%	68	100.00%			

**Table 8 ijms-21-01765-t008:** Minimum allelic frequency (MAF) in Sardinian population of single nucleotide polymorphisms (SNPs) under investigation. WNT4, wingless-typemammalian mouse tumour virus integration site family member 4; VEZT, vezatin; FSHB, follicle stimulating hormone beta polypeptide.

Gene	SNP	MAF Sardinia
WNT4	rs7521902	0.0971
VEZT	rs10859871	0.407
FSHB	rs11031006	0.106

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
