# Peer review of "Genetic Characterization of Endometriosis Patients: Review of the Literature and a Prospective Cohort Study on a Mediterranean Population"

_ijms, 2020, doi:10.3390/ijms21051765_

Round 1

Reviewer 1 Report

This is an interesting paper from a well-known group. The genetic/epigenetic research about endometriosis should be encouraged and this review may be helpful. However, additional clarifications are needed in order to better interpret the findings of this study.

First of all, you should better specify the nature of your study because I think that there are some incongruities. A case-control study is a retrospective study by definition: analysing material and methods section it seems to be unclear. Is this a prospective study? If you enrolled women before surgery, the study should be described as prospective. Moreover, in case of a case-control study you should enroll in the control group the same number of patients of the “case” group, at least.

ABSTRACT

Line 23-24: In material and methods section you should expose indications about the “review” portion of your paper.

Line 48-49: “in an attempt to define…” please restate. It is not clear.

INTRODUCTION

Well written. The historical part is very interesting but too long. On the other hand, there are no data about the specific genes you are evaluating in your paper. Please, reduce the historical part and discuss Literature data about polymorphism of genes.

MATERIALS AND METHODS

Line 98-104: Did you evaluate only English language study? Please specify.

Line 127-129: Nowadays can Sardinia be still defined as a genetic macro-isolate? Do you have some previous evidences in Literature showing that enrolling patient “with Sardinian origins for at least 3 generations” is enough to define patients with Sardinian origin? Please add references of studies with similar setting.

Line 142-146: It is not clear if both case and control groups underwent surgery. Did you have histological confirmation for both group? It seems that only control group had surgery: in this case please add the absence of histological confirmation of “case group” as a limitation of the study.

RESULTS

Line 269 substitute “too” with “also”

Line 343 Please restate the sentence “Attar et al in 2010 reported…”

Table 2 did you perform statistics on age, bmi etc? report p value in the table.

Table 3 please report in the table how you define “mild/moderate/severe” for endometriosis symptoms (e.g. VAS 0-3 vs 4-7 vs 8-10)

Table 6, Table 7 and Table 8 have extra lines above seventh/eighth columns.

Figure 1,Figure 2 and Figure 3: Under “phenotype” case and control are in Italian language as Frequenza on the left part of the figure. Moreover, you should enlarge the font size for all the written part of the figures.

DISCUSSION AND CONCLUSIONS

Well written.

Reviewer 2 Report

This work describes the effect of genetic variants on the development and progress of endometriosis, using literature references and a control study. There is a great interest around this topic of research and it is important that the new findings are clearly represented. Authors should be more specific on how their results contribute to what is already known and how they evaluate their results.

It is known that a number of pathological conditions (Crohn's disease, type 1 diabetes and infammatory bowel disease) have been associated with endometriosis (DOI:10.1038/s41598-018-29462-y, DOI: 10.3892/mmr.2018.9521). Therefore, the relevant pathological medical history of some participants in the control group, used for the control study, raise some doubts about the appropriateness of this sample.

I would suggest a native English speaker to correct syntax errors.

Authors need to give more info about their literature search: the number of initial articles, the inclusion and exclusion criteria used.

Minor corrections 

Please, use L instead of l as the unit of volume (ex mL not ml)

Table 1, 2, 6, 7, 8: use dots for decimal places

Tables: You need to add the title of each table at the top of the table

Figure 1, 2, 3: use English

Figure 4: hormonal not ormonal

 Please, mind consistency: for example either 25μL OR 25 μL

29 VEZT and FSHB genes was evaluated in patients and controls. In this work, is described the

30 frequency of the alleles, genotypes and haplotypes of these SNPs were described in Sardinian women.

54 Endometriosis is a chronic disease

56 multifactorial aetiology in endometriosis, with possible causes in genetic

60 studies on genetics genetic characteristics

63 comorbidities capable of generating bias [3].In the field of studies research

71 population, which not only confirmed on a large scale the results of previous studies on a large scale

74 affected), but also highlighted how  the highest higher degree of familiarity was detected in those cases of

75 endometriosis with a heavier severity of severe symptoms 

90 endometriosis affected women and to evaluate the influence of the polymorphisms of genes WNT4,

91 VEZT, FSHB, known to be involved in molecular mechanisms associated with a phenomena of

147 geographical origin ad and anthropometric

169 9ml mL EDTA tube) for genetic analysis.

240 sex hormones: oestrogens and progesterone [30-31]. On this subtle mechanism the choice of the

241 candidate genes for various association studies has fallen is based on this mechanism.

246 but with inconsistent results. In For ESR2 gene

253 in the development of endometriosis is based. In particular, object of numerous studies also on the their

258 from endometriosis. However, also this affirmation finds important differences and contrasting

259 results regarding different ethnic groups. (It is not clear what you mean. Please, rephrase.)

307 VEGF (Vascular Endothelial Growth Factor) … You need to explain the abbreviation the first time that appears in the text and this is in line 283

380 obsolete, opening the way to the more promising genome-wide association studies (GWAS). A

381 genome-wide association study (GWAS) … you have already explained the abbreviation above …

552 568 600 614 617 a full stop missing at the end of the sentence

563 a parenthesis missing

575 GA and the polymorphic minor allele A) (Table 6 e and Figure 1).

621 Nevertheless, although it seems is clear that there are differences in genetic associations with

622 in endometriosis among between the different populations of the world.

Reviewer 3 Report

Nicely organised study.could be accepted for publication,however needs minor linguistic revision.

Author Response

Response to Reviewer 3 Comments

Point 1: Nicely organised study.could be accepted for publication,however needs minor linguistic revision.

 Response 1: Thanks for your revision. We have submitted our manuscript for English language corrections via https://www.mdpi.com/authors/english ( edited 16494)

Round 2

Reviewer 1 Report

The corrections that you made are correct.